# OpenReview forum: "Zero-Shot Learning of Causal Models"
_ICLR.cc/2025/Conference — Submitted to ICLR 2025_

### Official Review · Reviewer_gTiX · 2024-11-03

**Soundness:** 3
**Presentation:** 3
**Contribution:** 3
**Rating:** 6
**Confidence:** 3

**Summary:**

This paper proposes a method to learn a single model capable of inferring the causal generative processes of datasets in a zero-shot manner, rather than learning a specific model for each dataset. The general idea is to amortize the learning of a conditional version of the FiP architecture, which can infer the generative SCMs directly from observations and causal structures on synthetically generated datasets.

**Strengths:**

Zero-shot learning of a causal model is an important task, potentially due to the limitation of the training dataset when training the model. So I believe the task proposed by the paper is highly motivated. Additionally, the paper is written in a structured way.

**Weaknesses:**

1. I would suggest author to either use a figure or introduce a bit more background of the idea of amortized causal learning in related works.

2. In background on causal learning, I still don't know how to compute H. and what $Jac_x$ is. Also, $d_n$ and M are not defined.

3. In training setting, I don't understand how to use noise samples. In equation (1), they seem to be residual terms but in Section 3.1, they are said to play the rule of the target variable.

4. Do we need specific requirements on training data? Are they supposed to cover as many as domains?

5. Why modeling noise will help zero-shot ability of SCM? I wish the paper gives more explicit explanations.

6. In Table 2 and Table 3, it seems that sometimes Cond-FiP has the worse performance than FiP. This needs to be explained.

**Questions:**

Please see weakness above.

**Details Of Ethics Concerns:**

No ethics concerns.

---

> ### Author Response · Authors · 2024-11-22
> **Rebuttal by Authors: Part 1/2**
>
> > I would suggest author to either use a figure or introduce a bit more background of the idea of amortized causal learning in related works.
>
> Thank you for this comment. We have added some precisions on amortizated learning at the begining of the rebuttal to explain the usage of such techniques in causal ML and we will update the background section of our manuscript accordingly. We hope that these clarifications adequately address the concerns of the reviewer.
>
> > In background on causal learning, I still don't know how to compute $H$. and what $Jac_x$ is. Also,  $d_n$ and $M$ are not defined.
>
> Thanks for pointing out these imprecisions. We will clarify these notations in the background section.
>
> By denoting $\textbf{F}:=[F_1,\dots,F_d]$, and $P\in\{0,1\}^{d\times d}$ the permutation matrix associated to a topological order, i.e. $P_{i,j}=1$ i $\sigma(i)=j$ and $0$ otherwise where $\sigma$ is a topological ordering (TO), then $\textbf{H}(X,N):=\textbf{F}(PX, PN)$.
>
> $Jac$ is the notation we adopt from FiP to denote the Jacobian, where the subscript indicates the Jacobian with respect to the first variable (x) or the second variable (n).
>
> $d_h$ is the hidden dimension of the queries and keys $Q$ and $K$ respectively in the attention.
>
> Finally, $M$ is a mask matrix that satisfies $M_{i,j}=+\infty$ if $(i,j)$ is masked, and $0$ otherwise.
>
> > In training setting, I don't understand how to use noise samples. In equation (1), they seem to be residual terms but in Section 3.1, they are said to play the rule of the target variable.
>
> In SCM, the noise are the exogenous variables from which the observable ones are generated according to some functional mechanisms. More formally, for any sample noise $N$, there exists a unique observable sample $X$ associated according to a causal functional mechanism $\textbf{F}$. Therefore given an observation $X$, the task of recovering the noise sample associated is equivalent to learn the inverse generartive process, which is (under inversibility assumption such as in ANM) equivalent to learn the generative process itself. Therefore, recovering noise is an excellent task to learn an embedding of the datasets. We will clarify this point in our draft.
>
> > Do we need specific requirements on training data? Are they supposed to cover as many as domains?
>
> Our training requirements, as specified in the manuscript, involve the knowledge of noise samples and causal graphs to learn our amortized method. However, once trained, the model can be applied to any type of data without requiring access to noise samples. We investigate the generalization capabilities of our approach by testing the learned model on out-of-distribution problems and observe that the model manages to recover the performance of SoTA baselines trained specifically for these new datasets. To further enhance the generalization capabilities of our approach, we would need to scale both the model and the training data, allowing the model to encounter more complex and diverse SCMs. We leave this investigation for future work.
>
>
> > Why modeling noise will help zero-shot ability of SCM? I wish the paper gives more explicit explanations.
>
> Thank for you asking this important question. As we mentioned earlier, recovering the noise from data is equivalent to learn the inverse causal generative process. Being able to recover this process is a certificate that the model has understood how the data has been generated. We will clarify this point in our manuscript.

---

> > ### Comment · Reviewer_gTiX · 2024-11-25
> > **Thanks for the response**
> >
> > I think authors have partially addressed my concerns. For Q4 it seems that the author does not answer directly. For Q3, I would suggest the author to insert a theory to the paper to mathematically present how recovering the noise is equivalent to learn the inverse causal generative process. I'd like to increase my score as most of my concerns have been addressed.

---

> ### Author Response · Authors · 2024-11-22
> **Rebuttal by Authors: Part 2/2**
>
> > In Table 2 and Table 3, it seems that sometimes Cond-FiP has the worse performance than FiP. This needs to be explained.
>
> Recall that Cond-FiP infers in zero-shot the SCMs and therefore, the parameters of the model does not change when a new dataset is given as input. However, FiP is trained from scratch for each new instance problem, or dataset it sees. Therefore FiP performances can be seen as the gold standard that Cond-FiP need to reach in order to be competitive, and we would typically not expect Cond-FiP to improve upon FiP in term of performance. However, Cond-FiP is able to predict the SCM in zero-shot, while FiP requires to train the model from scratch.
>
> Also, in the general response above, we have added new experiments comparing Cond-FiP with other baselines in a scarce data regime. **In such a scenario, we show that Cond-FiP obtains significant improvement over FiP (and others)**. Indeed, when the input context size is small we cannot reliably train models from scratch. In contrast, zero-shot inference methods are less impacted as their parameters remain unchanged when given a new dataset, and the inductive bias they have learned during their training phase enables them to generalize even with a smaller input context. More details on these experiments can also be found in Appendix D.2.
>
> For the convenience of the reader, we state the results again from the general response above comparing Cond-FiP against FiP.
>
> ### Experiments with $n_{\text{test}}=100$
>
> | Method   | Total Nodes | LIN | RIN | LOUT | ROUT |
> |----------|-------------|----------------|----------------|-----------------|-----------------|
> | FiP     | 10          | 0.13 (0.01)    | 0.29 (0.04)    | 0.18 (0.02)     | 0.15 (0.03)     |
> | Cond-FiP | 10          | 0.09 (0.01)    | 0.20 (0.03)    | 0.09 (0.02)     | 0.14 (0.02)     |
> | FiP     | 20          | 0.17 (0.02)    | 0.34 (0.06)    | 0.17 (0.02)     | 0.39 (0.03)     |
> | Cond-FiP | 20          | 0.08 (0.00)    | 0.31 (0.06)    | 0.13 (0.01)     | 0.37 (0.02)     |
> | FiP     | 50          | 0.13 (0.01)    | 0.38 (0.07)    | 0.14 (0.01)     | 0.58 (0.06)     |
> | Cond-FiP | 50          | 0.10 (0.01)    | 0.32 (0.05)    | 0.12 (0.01)     | 0.54 (0.05)     |
> | FiP     | 100         | 0.15 (0.01)    | 0.40 (0.07)    | 0.19 (0.02)     | 0.67 (0.07)     |
> | Cond-FiP | 100         | 0.11 (0.01)    | 0.35 (0.07)    | 0.14 (0.02)     | 0.63 (0.07)     |
>
> ### Experiments with $n_{\text{test}}=50$
>
>
> | Method   | Total Nodes | LIN | RIN | LOUT | ROUT |
> |----------|-------------|----------------|----------------|-----------------|-----------------|
> | FiP     | 10          | 0.13 (0.03)    | 0.27 (0.03)    | 0.15 (0.02)     | 0.21 (0.03)     |
> | Cond-FiP | 10          | 0.09 (0.01)    | 0.17 (0.01)    | 0.11 (0.01)     | 0.16 (0.01)     |
> | FiP     | 20          | 0.12 (0.01)    | 0.33 (0.04)    | 0.15 (0.02)     | 0.35 (0.04)     |
> | Cond-FiP | 20          | 0.10 (0.01)    | 0.16 (0.01)    | 0.11 (0.01)     | 0.17 (0.01)     |
> | FiP     | 50          | 0.13 (0.01)    | 0.32 (0.03)    | 0.13 (0.01)     | 0.49 (0.05)     |
> | Cond-FiP | 50          | 0.10 (0.01)    | 0.16 (0.00)    | 0.10 (0.01)     | 0.17 (0.01)     |
> | FiP     | 100         | 0.11 (0.01)    | 0.32 (0.04)    | 0.13 (0.01)     | 0.48 (0.02)     |
> | Cond-FiP | 100         | 0.09 (0.01)    | 0.16 (0.01)    | 0.09 (0.01)     | 0.18 (0.01)     |
>
> Hence, in these scarce data regimes, Cond-FiP shows significant improvements compared to baselines that require to be trained from scratch and so for each test dataset.

---

> ### Author Response · Authors · 2024-11-26
> **Many thanks for reading our rebuttal**
>
> Dear Reviewer,
>
>
>
> Many thanks for reacting to our rebuttal. We are grateful that you have raised your score despite your remaining concerns.
>
> Let us now clarify further the points you mentioned.
>
>
> > For Q3, I would suggest the author to insert a theory to the paper to mathematically present how recovering the noise is equivalent to learn the inverse causal generative process.
>
>
>
> We will follow your suggestion and add the following paragraph to the manuscript to clarify further the training of the encoder.
>
>
> Recall that an SCM is an **implicit** generative model that, given a noise sample $\textbf{N}$, generates the corresponding observation according to the following fixed-point equation in $\textbf{X}$
>
>  $\textbf{X}=F(\textbf{X},\textbf{N})$
>
> More precisely, to generate the associated observation, one must solve the above fixed-point equation in $\textbf{X}$ given the noise $\textbf{N}$.
>
> Let us now introduce the following notation that will be instrumental for the subsequent discussion: we denote $F_{\textbf{N}}(z): z\to F(z, \textbf{N})$.
>
> Due to the specific structure of $F$ (determined by the DAG $\mathcal{G}$ associated with the SCM), the fixed-point equation mentioned above can be efficiently solved by iteratively applying the function $F_{\textbf{N}}$ to the noise (see Eq. (5) in the manuscript). As a direct consequence, the observation $\textbf{X}$ can be expressed as a function of the noise:
>
> $\textbf{X} = F_{\text{gen}}(\textbf{N})$
>
>
> where $F_{\text{gen}}(\textbf{N}):= (F_{\textbf{N}})^{\circ d}(\textbf{N})$, $d$ is the number of nodes, and $\circ$ denotes the composition operation. In the following we refer to $F_{\text{gen}}$ as the **explicit** generative model induced by the SCM.
>
> Conversely, assuming that the mapping $z\to F_{\text{gen}}(z)$ is invertible, then one can express the noise as a function of the data:
>
> $\textbf{N} = F_{\text{gen}}^{-1}(\textbf{X})$
>
>
> Therefore, learning to recover the noise from observation is equivalent to learn the function $F_{\text{gen}}^{-1}$, which is exactly the inverse of the explicit generative model  $F_{\text{gen}}$.
>
> It is also worth noting that under the ANM assumption (i.e. $F(\textbf{X},\textbf{N})=f(\textbf{X}) + \textbf{N}$), $F_{\text{gen}}$ is in fact always invertible and its inverse admits a simple expression which is
>
> $F_{\text{gen}}^{-1}(z)  = z - f(z)$.
>
> Therefore, in this specific case, learning the inverse generative model  $F_{\text{gen}}^{-1}$ is exactly equivalent to learning the causal mechanism function $f$.
>
>
> Let us now provide a more detailed and precise response to Q4.
>
> > Do we need specific requirements on training data?
>
> To train our current model, it is essential to consider a substantial number of datasets. Specifically, we have trained Cond-FiP using 4 million synthetically SCMs, as detailed in line 377.
>
>
> Another important consideration is that all these SCMs are sampled according to the same distribution ($P_{\textbf{IN}}$ as defined in paragraph l.355). The complexity of the SCM distribution directly influences the complexity required for the model. In our current setting, we observe that a model with 20 million parameters is capable of effectively handling such a distribution of SCMs, as it generalizes well on new test datasets generated from $P_{\textbf{IN}}$ (refer to results in Tables 1, 2, and 3 for LIN **IN** and RFF **IN**). Additionally, we examine the generalization capabilities of our approach by testing the trained model on SCM out-of-distribution problems (LIN **OUT** and RFF **OUT**) sampled according to $P_{\textbf{OUT}}$, a shifted version of $P_{\textbf{IN}}$. Our observations indicate that the model successfully recovers the performance of SoTA baselines specifically trained for these new datasets.
>
>
>
> Finally, as mentioned in our previous response, the knowledge of noise samples and causal graphs is necessary for training our amortized method. However, during inference, only access to observational data is required.
>
>
> > Are they supposed to cover as many as domains?
>
> Currently, the SCM distribution considered during training ($P_{\textbf{IN}}$) encompasses a non-trivial set of possible linear and non-linear SCMs with various types of graphs and noise. However, there remains significant potential for enhancing the generalization capabilities of our model by increasing the complexity of the SCM distribution. This would necessitate scaling the model accordingly to accommodate the learning of more complex and diverse SCMs. We leave this investigation for future work.
>
>
>
>
> We hope that these clarifications adequately address your concerns.
>
> Thanks again for reading our rebuttal, and for your detailed review.

---

### Official Review · Reviewer_ADS4 · 2024-11-03

**Soundness:** 3
**Presentation:** 4
**Contribution:** 3
**Rating:** 5
**Confidence:** 3

**Summary:**

The paper proposes to amortize the learning of a conditional version of FiP to infer directly the generative SCMs from observations and causal structures in a zero-shot manner.

**Strengths:**

1.The idea of zero-shot learning with conditional FiP is interesting.

2.The problem studied is important in the sense that zero-shot learners are commonly needed to finish the tasks.

**Weaknesses:**

1.The theoretical  justifications need to be improved.
2. Some experiments are missing.

**Questions:**

1.The theoretical foundation of "ZERO-SHOT LEARNING OF CAUSAL MODELS" indeed needs to be improved, given its complexity and the challenges it presents in the field of machine learning. Because the cross domain SCM difference seems to be significant. I still feel puzzled why zero shot learning theoretically works. Is there problems such as incapability of the observation-based learning with conditional FiP?

2.In the experimental section, I think more comparisons are needed. For example in DoWHY dataset, you may vary the node number and more generative mechanisms. Also, some ablations study of the important modules of FiP (e.g., encoder) should be presented since this seems to be an important factor for the final results.

3.I still feel that the interventional and sample generation can be more comprehensive. Is there any demo example to show that how your method differs from others? I guess that which node you manipulate still has a huge impact on the result. Am I right or not?

---

> ### Author Response · Authors · 2024-11-22
> **Rebuttal by Authors**
>
> > The theoretical foundation of "ZERO-SHOT LEARNING OF CAUSAL MODELS" indeed needs to be improved, given its complexity and the challenges it presents in the field of machine learning. Because the cross domain SCM difference seems to be significant.
>
> Thank you for this comment. We have added some precisions on amortizated learning at the begining of the rebuttal to explain the usage of such techniques in causal ML and we will update the background section of our manuscript accordingly. We hope that these clarifications adequately address the concerns of the reviewer.
>
> >I still feel puzzled why zero shot learning theoretically works. Is there problems such as incapability of the observation-based learning with conditional FiP?
>
> In order to increase the capacity of inferring more diverse functions, we would need to scale the model further, and generates even more complex and various synthetic data, but we consider this matter out of the scope of this paper.
>
>
> > In the experimental section, I think more comparisons are needed. For example in DoWHY dataset, you may vary the node number and more generative mechanisms.
>
> Thank you for this remark. Concerning DoWhy,
> we use the automatic assignment selection offered by the library with quality sets to *AssignmentQuality.GOOD* to learn SCMs. We also want to precise that we vary the number of nodes, the noise, graphs and the function distributions in all our experiments to evaluate the performances of Cond-FiP.
>
>
> > Also, some ablations study of the important modules of FiP (e.g., encoder) should be presented since this seems to be an important factor for the final results.
>
> We thank the reviewer for their suggestion!  We have conducted an ablation study for the encoder, please refer to the general response above and Appendix D.3 for more details. We hope that these additional experiments address the concerns of the reviewer.
>
> > I still feel that the interventional and sample generation can be more comprehensive. Is there any demo example to show that how your method differs from others?
>
> Thank you for highlighting this point. We have included detailed explanations on the usage of Cond-FiP at inference time for both generation and interventional prediction tasks in the general response. We hope that these clarifications adequately address the concerns of the reviewer.
>
>
> > I guess that which node you manipulate still has a huge impact on the result. Am I right or not?
>
> The reviewer is absolutely correct. Certain interventions are more challenging to estimate than others. Therefore, to fairly evaluate the capabilities of both the baselines and our approach, we randomly sample intervened nodes and values according to the full graph with uniform probability and average the predictions over multiple random interventions and samples. We will clarify this point in the manuscript.

---

> > ### Comment · Reviewer_ADS4 · 2024-11-26
> >
> > Thank you for the responses. Some of my concerns are cleared, and I would like to keep my score unchanged.

---

> > > ### Author Response · Authors · 2024-11-26
> > > **Thanks for reading our rebuttal**
> > >
> > > Dear Reviewer,
> > >
> > > thank you for reacting to our rebuttal.
> > >
> > > We are pleased to hear that we have addressed some of your concerns. However, it appears that there are still some points that need clarification. We would greatly appreciate any further suggestions you may have to help us improve our work.
> > >
> > > Thanks again for reading our rebuttal, and for your review.

---

> > > > ### Author Response · Authors · 2024-11-30
> > > > **Kind Reminder**
> > > >
> > > > Dear Reviewer,
> > > >
> > > > Thank you once again for reviewing our rebuttal and for your prompt response. We would be more than happy to address any remaining concerns you may have.
> > > >
> > > > We are also very pleased that you rated our manuscript with a score of **3** for **Soundness**, **4** for **Presentation**, and **3** for **Contribution**. However, we noticed that your overall rating is **5**, which does not seem to reflect the individual scores you provided.
> > > >
> > > > Please let us know if there are any specific points in our work that you feel need further improvement. We would be grateful for your additional suggestions and are eager to make the necessary revisions.
> > > >
> > > > Thank you again for your review and your time.

---

### Official Review · Reviewer_iSSj · 2024-11-04

**Soundness:** 2
**Presentation:** 3
**Contribution:** 2
**Rating:** 6
**Confidence:** 3

**Summary:**

The authors propose a novel method and task of inferring functional relationships of SCMs from observations and their associated causal structures in a zero-shot manner. The problem is well motivated by the literature. The main technical contributions of the work are the definition of Cond-FIP (modification to the FIP architecture), and the construction of the dataset embeddings (used to condition FIP). The latter is done by training the transformer-based encoder to predict the evaluations of the functional relationships over the distribution of SCMs. The former is done using the FIP architecture that takes as input sample embeddings conditioned on the SCM and is trained with the MSE objective on samples from multiple SCMs.

The method is evaluated in the synthetic setup on varying graph structures, functional relationships, and noises. It performs comparably to the selected set of baselines on noise prediction, sample generation, and interventional sample generation tasks.

**Strengths:**

1. The problem is well motivated by recent advances in the area, novel, and of interest to the community.
2. The method’s description is detailed and well-structured.

**Weaknesses:**

1. Experimental evaluation seems limited. The authors claim that their approach matches the performance of SoTA, but they only compare against a small set of methods. It seems natural that authors would compare against cited work (Javaloy et al., 2023; Khemakhem et al., 2021) and architectures used in causal discovery literature [1, 2],
2. In various fragments of the text the authors claim that “this is the first time that SCMs are inferred in a zero-shot manner from observations“. While it is clear to me that this work combined with previous literature technically allows SCM inference (graph and functional relationships), this ability is not demonstrated nor evaluated in this work.
3. The method does not generalize well to larger datasets (as stated in Appendix C). This is an important limitation and should be stated explicitly in the main text.
4. The evaluation would be hard to reproduce as the details of train and test dataset generation are not provided.
5. Some parts of the method’s explanation are unclear to me. (see Questions)

[1] Annadani at al., BayesDAG: Gradient-Based Posterior Inference for Causal Discovery, NeurIPS 2023
[2] Brouillard et al., Differentiable Causal Discovery from Interventional Data, NeurIPS 2020

**Questions:**

1. I have a hard time understanding the Generation with Cond-FiP paragraph. The \mathcal{T} symbol seems to have a different meaning in the Projection paragraph than in the Generation with Cond-FiP. Is it a notational error? Please clarify.  Also, what is the relation between eq. 5 and the equation in the Adaptive Transformer Block paragraph? They seem oddly similar yet, if I understand correctly, the \mathcal{T} denotes the whole generation process described in the Adaptive Transformer Block paragraph.
2. The Interventional Predictions explanation is unclear to me. Could you elaborate more on how you “modify in place the SCM obtained by Cond-FiP”?
3. Could you provide more details on the generation of the datasets? What is the density of the graphs used in training and evaluation? What are the differences in the parametrization of functional relationships between P_in and P_out?
4. What kind of regressor was used as a causal mechanism in DoWhy? What architecture was used in DECI?
5. What is the role of DAG attention in dataset embedding? Can you elaborate on what would happen if you were to use standard dot product attention in the dataset encoding phase?
6. It seems that the performance of baselines on RFF OUT dataset degrades significantly compared to RFF IN. Could you explain why?

---

> ### Author Response · Authors · 2024-11-22
> **Rebuttal by Authors: Part 1/3**
>
> > Experimental evaluation seems limited. The authors claim that their approach matches the performance of SoTA, but they only compare against a small set of methods. It seems natural that authors would compare against cited work (Javaloy et al., 2023; Khemakhem et al., 2021) and architectures used in causal discovery literature*
>
> **Update**: Please check the part 3 of our response below, where we have provided results with the CausalNF (Javaloy et al., 2023) baseline.
>
> Thank you for raising this point. We are currently benchmarking Cond-FiP against the additional baselines suggested by the reviewer. Regarding prior works (Javaloy et al., 2023; Khemakhem et al., 2021), a benchmark has already been conducted in FiP, and the authors found that FiP's performance is on par with, or even superior to, these approaches. Therefore, our current results on benchmarking against FiP already provide a strong indication of the performance of our proposed approach.
>
> > In various fragments of the text the authors claim that “this is the first time that SCMs are inferred in a zero-shot manner from observations“. While it is clear to me that this work combined with previous literature technically allows SCM inference (graph and functional relationships), this ability is not demonstrated nor evaluated in this work.
>
> Thank you for acknowledging that the combination of prior work with our framework would enable SCM inference. To further demonstrate this, we have conducted additional experiments showing the extension of our work when the causal graphs are unknown at inference time. To achieve this, we leverage prior work on amortized causal graph learning (AVICI) to predict the graphs in a zero-shot manner from observations only.
>
>  **We find that Cond-FiP remains competitive to baselines in this scenario of no access to true causal graph knowledge!** Therefore, our amortized training procedure can be seamlessly adapted for zero-shot inference of both causal graphs and causal mechanisms from observations only. Please refer to the general response above, and Appendix D.1 for more details.
>
> > The method does not generalize well to larger datasets (as stated in Appendix C). This is an important limitation and should be stated explicitly in the main text.
>
> We concur with the reviewer and have updated our manuscript to clarify this limitation (see paragraph on limitations p. 9). The observed phenomenon occurs because our current model is biased and limited by the training setting. Specifically, the model has learned to generate functions given $n_{\text{train}}=400$ samples. Consequently, increasing the context size at inference time does not result in a significant improvement in our approach. To enhance predictions for larger input contexts, we would need to train the model accordingly, which requires considerably more resources. We leave this investigation for future work.
>
>
> > I have a hard time understanding the Generation with Cond-FiP paragraph. The \mathcal{T} symbol seems to have a different meaning in the Projection paragraph than in the Generation with Cond-FiP. Is it a notational error? Please clarify.*
>
> Given a dataset $D_X$ and a graph $\mathcal{G}$, we denote $z\to\mathcal{T}(z,D_X,\mathcal{G})$ the function inferred by our neural network as shown in Figure 1. This function defines the predicted SCM obtained by Cond-FiP that can be used to generate new points as explained in Eq. (5). We have also included detailed explanations on the usage of Cond-FiP at inference time for both generation and interventional prediction tasks in the general response above. We hope that these clarifications adequately address the concerns of the reviewer.
>
>
> > Also, what is the relation between eq. 5 and the equation in the Adaptive Transformer Block paragraph? They seem oddly similar yet, if I understand correctly, the \mathcal{T} denotes the whole generation process described in the Adaptive Transformer Block paragraph.*
>
> Thank you for raising this point. Indeed, these two processes are similar because the adaptive transformer block aims to replicate the generation process. One can consider the adaptive transformer block as the unfolded version of the generation process, parameterized in a differentiable manner to be learned from data. We will clarify this point in the manuscript.
>
> > What kind of regressor was used as a causal mechanism in DoWhy? What architecture was used in DECI?*
>
> For DoWhy we use the automatic assignment selection offered by the library with quality sets to *AssignmentQuality.GOOD*. For DECI we use the default implementation with Gaussian noise.

---

> ### Author Response · Authors · 2024-11-22
> **Rebuttal by Authors: Part 2/3**
>
> > What is the role of DAG attention in dataset embedding? Can you elaborate on what would happen if you were to use standard dot product attention in the dataset encoding phase?
>
> DAG attention enables the modeling of functional relationships in SCMs. Given an SCM $(x,n) \to F(x,n)$, and considering the nodes ranked in the topological order, the Jacobian $Jac_{x} F$ obtained must be strictly lower triangular. This is because, when reordered, the nodes exhibit strict autoregressive relationships. Additionally, the root nodes have no connections with other nodes. Consequently, standard attention mechanisms are inadequate for modeling such graphs, necessitating the use of DAG attention to accurately model SCMs. We will precise this point in our manuscript.
>
> > Could you provide more details on the generation of the datasets? The evaluation would be hard to reproduce as the details of train and test dataset generation are not provided. What is the density of the graphs used in training and evaluation? What are the differences in the parametrization of functional relationships between P_in and P_out?*
>
> We agree with the reviewer that some details regarding dataset generation were not provided and we will clarify this in our manuscript. To generate the synthetic data, we use the exact same setup as AVICI for synthetic data generation, please check the Table 3 in their Appendix (https://arxiv.org/pdf/2205.12934) where they precisely define the setting for both in and out distribution. More precisely, the sampled graphs can take edge density per node from the following posssible values: $[1, 2, 3]$, and can follow multiple graph distributions such as Erdos-Rényi,  Scale-free,  Watts-Strogatz, and  Stochastic block model. For the parameterization of functional relationships, in general the support over the parameters in the out-of-distribution setting is significantly larger than the one considered at training time.
>
>
> > The Interventional Predictions explanation is unclear to me. Could you elaborate more on how you “modify in place the SCM obtained by Cond-FiP”?
>
> Thank you for highlighting this point. We have included detailed explanations on the usage of Cond-FiP at inference time for both generation and interventional prediction tasks in the general response above. We hope that these clarifications adequately address the concerns of the reviewer.
>
> > It seems that the performance of baselines on RFF OUT dataset degrades significantly compared to RFF IN. Could you explain why?
>
> The primary reason is that the distribution shift in the non-linear setting is more pronounced than in the linear case. Specifically, the non-linear functional mechanisms considered are non-trivial expansions of 100 functions, each of which can be shifted with the same magnitude as in the linear case, which consists of a single linear function.

---

> ### Author Response · Authors · 2024-11-26
> **Rebuttal by Authors: Part 3/3**
>
> ### Additional Benchmark
>
> We are delighted to compare our method with CausalNF (Javaloy et al., 2023) and appreciate the reviewer's suggestion, as it enhances the comprehensiveness of our baseline comparisons.
>
> The results for the task of noise prediction and sample generation are presented below. The test datasets consist of $n_{\text{test}}= 400$ samples, exact same setup as in our main results (Table 1, 2, 3). To ensure a fair comparison, we provided CausalNF with the true causal graph.
>
> ## Noise Prediction
>
> | Method   | Total Nodes | LIN   | RIN   | LOUT  | ROUT  |
> |----------|-------------|-------|-------|-------|-------|
> | CausalNF | 10          | 0.16 (0.02) | 0.41 (0.09) | 0.38 (0.04) | 0.35 (0.02) |
> | Cond-FiP  | 10          | 0.06 (0.01) | 0.10 (0.01) | 0.07 (0.01) | 0.10 (0.01) |
> | CausalNF | 20          | 0.18 (0.03) | 0.45 (0.12) | 0.29 (0.05) | 0.36 (0.03) |
> | Cond-FiP  | 20          | 0.06 (0.01) | 0.09 (0.01) | 0.07 (0.00) | 0.12 (0.00) |
> | CausalNF | 50          | 0.25 (0.03) | 0.56 (0.09) | 0.45 (0.06) | 0.38 (0.04) |
> | Cond-FiP  | 50          | 0.06 (0.01) | 0.10 (0.01) | 0.07 (0.01) | 0.14 (0.01) |
> | CausalNF | 100         | 0.24 (0.02) | 0.80 (0.1)  | 0.37 (0.06) | 0.49 (0.05) |
> | Cond-FiP  | 100         | 0.05 (0.0)  | 0.10 (0.01) | 0.07 (0.01) | 0.16 (0.01) |
>
> ## Sample Generation
>
> | Method   | Total Nodes | LIN   | RIN   | LOUT  | ROUT  |
> |----------|-------------|-------|-------|-------|-------|
> | CausalNF | 10          | 0.27 (0.07) | 0.29 (0.04) | 0.20 (0.03) | 0.20 (0.03) |
> | Cond-FiP  | 10          | 0.06 (0.01) | 0.14 (0.02) | 0.05 (0.01) | 0.08 (0.01) |
> | CausalNF | 20          | 0.23 (0.02) | 0.36 (0.05) | 0.22 (0.02) | 0.45 (0.02) |
> | Cond-FiP  | 20          | 0.05 (0.01) | 0.24 (0.06) | 0.07 (0.01) | 0.30 (0.03) |
> | CausalNF | 50          | 1.5 (0.26) | 0.93 (0.13) | 3.09 (0.55) | 0.95 (0.04) |
> | Cond-FiP  | 50          | 0.08 (0.01) | 0.25 (0.05) | 0.07 (0.00) | 0.48 (0.07) |
> | CausalNF | 100         | 1.23 (0.13) | 0.85 (0.08) | 1.67 (0.13) | 0.96 (0.04) |
> | Cond-FiP  | 100         | 0.07 (0.01) | 0.29 (0.07) | 0.09 (0.01) | 0.57 (0.07) |
>
> Our analysis reveals that CausalNF underperforms compared to Cond-FiP in both tasks, and it is also a weaker baseline relative to FiP. Note also the authors did not experiment with large graphs for CausalNF; the largest graph they used contained approximately 10 nodes. Also, they trained CausalNF on much larger datasets with a sample size of 20k, while our setup has datasets with 400 samples only.
>
> We believe that we have thoroughly addressed all the concerns raised by the reviewer. We are grateful for the insightful suggestions and appreciate the opportunity to clarify our work. As the rebuttal period draws to a close, we would like to confirm whether our responses have adequately addressed all the concerns.

---

> > ### Comment · Reviewer_iSSj · 2024-11-29
> >
> > Thank you for providing the additional experimental results and for your detailed responses. I appreciate the effort in addressing my concerns. My concerns have been resolved, and I am happy to increase my score.

---

> > > ### Author Response · Authors · 2024-11-30
> > > **Many thanks for reading our rebuttal**
> > >
> > > Dear Reviewer,
> > >
> > > Thank you for your insightful comments and questions. They have significantly enhanced our manuscript. We are also very thankful for your score increase.

---

### Official Review · Reviewer_6M7T · 2024-11-05

**Soundness:** 3
**Presentation:** 3
**Contribution:** 3
**Rating:** 8
**Confidence:** 3

**Summary:**

Learning the causal generative process from observational data is a challenging problem bottlenecked by the necessity of learning a separate causal model for each dataset. This paper studies a unifying framework to enable zero-shot inference of causal generative processes of arbitrary datasets by training a single model. The authors adapt a recent advancement in causal generative modeling (FiP) to infer generative SCMs conditional on empirical dataset representations in a supervised setup, where the SCM is reformulated as a fixed-point problem. They propose an amortized procedure that takes in a dataset and its causal graph and learns a dataset representation. Then, the authors train a model conditioned on dataset embeddings to learn the functional mechanisms of the generative SCM. This framework enables both observational and interventional sample generation in a zero-shot manner. Empirical results show that the method performs competitively with baseline models for in-distribution and out-of-distribution settings.

**Strengths:**

- This paper is one of the first to consider generalizing the learning of functional mechanisms of structural causal models from arbitrary datasets and causal graphs and is a significant step toward building causally-aware foundation models.
- The paper is written well with clear intuitions and explanations as to how it relates to similar work (e.g., FiP).
- Although the assumptions are a bit strong (additive noise model, causal graphs known, noise variables known), the general idea of using an amortized procedure to approximate SCM distributions in a zero-shot manner is quite interesting.
- The empirical results are convincing and show interesting observations, especially the performance of sample generation under distribution shifts and as the causal graphs scale up. It is certainly impressive that Cond-FiP can approximate the SCM distribution of 50 and 100 graph nodes quite well given that it was trained on only datasets with 20-node graph size.

**Weaknesses:**

- In a synthetic data scenario, assuming access to the noise samples is a feasible assumption, but for real-world datasets, this will not typically hold. Using the noise samples as the supervision for the dataset embedding model may easily become unrealistic. The authors have an evaluation on a real-world benchmark (Sachs) in the appendix where they fit a Gaussian model. However, interventional sample results are not provided.
- I believe the idea to directly infer the SCM distribution under the additive noise model assumption is interesting. However, again, the feasibility of this assumption may not always hold. It is true that we often parameterize causal models as linear/nonlinear additive noise models, but this can be violated in practice. It seems that this approach would only hold under the strict assumption of additive noise models.
- Knowledge of the causal graph for several datasets can potentially be a strong assumption. In real-world datasets, the causal graph may be unknown and must be discovered. However, for the sake of this work, the synthetic scenarios are good for proof of concept.

**Questions:**

- Could the authors explain why Cond-FiP performs similar to some baselines in noise prediction and sample generation, especially when the node scale is the same or smaller than used in training? How is the FiP model implemented? In the original paper, it seems that the task is the recovery of the topological ordering. Is the FiP baseline here aware of the causal structure of datasets?
- How does the alternating application of transformer blocks E work? Is this just an alternating optimization method where you optimize for samples when nodes are fixed and optimize for nodes when samples are fixed?
- For zero-shot inference of the SCM distribution, what is the level of distribution shift that a new dataset can have for this method to be able to extrapolate well?
- The main capabilities of the proposed framework are noise prediction and observational/interventional sample generation. However, individual counterfactual sample generation is also important in many applications. Can this framework enable counterfactual inference?
- In the Adaptive Transformer Block, which iteratively updates the noise conditioned on the dataset embedding $z_{emb}$, can we interpret this as sort of a noise abduction typically performed in counterfactual inference?
- How exactly does one perform interventions in the Cond-FiP? It would be beneficial if the authors elaborate on this mechanism in the paper. From my understanding, we just feed in an intervened SCM causal graph with the mutilations and use the corresponding dataset embedding for conditional generation. However, this is not made very clear in the paper.

---

> ### Author Response · Authors · 2024-11-22
> **Rebuttal by Authors: Part 1/3**
>
> > In a synthetic data scenario, assuming access to the noise samples is a feasible assumption, but for real-world datasets, this will not typically hold. Using the noise samples as the supervision for the dataset embedding model may easily become unrealistic.
>
> Thank you for raising this point. We concur with the reviewer that most real-world problems do not provide such supervised signals. However, it is important to note that during inference, Cond-FiP does not require access to noise samples. Instead, it only needs the observations (and a predicted or true causal graph) to infer the functional mechanisms. Therefore we can still apply Cond-FiP on real-world datasets, as shown by our experiment on the Sachs dataset (Appendix A).
>
> Our proposed framework primarily aims to demonstrate the feasibility of inferring SCMs from their empirical representations. A very interesting extension of our work would be to consider a semi-supervised setting where synthetic data and real-world data are mixed at training time, but we believe this is out of the scope of this paper.
>
> > The authors have an evaluation on a real-world benchmark (Sachs) in the appendix where they fit a Gaussian model. However, interventional sample results are not provided.
>
> Regarding the lack of interventional generation results on Sachs, the main issue is that Cond-FiP (and the other baselines considered in this work) only supports hard interventions while the interventional data available for Sachs are soft ones (i.e. the interventional operations applied are unknown). As a result, we are unable to provide a comprehensive evaluation of Cond-FiP (as well as the other baselines) for interventional predictions on Sachs.
>
>
> >I believe the idea to directly infer the SCM distribution under the additive noise model assumption is interesting. However, again, the feasibility of this assumption may not always hold. It is true that we often parameterize causal models as linear/nonlinear additive noise models, but this can be violated in practice. It seems that this approach would only hold under the strict assumption of additive noise models.
>
> Thank you for acknowledging the soundness of our framework. Our method relies on the ANM assumption only for the training the encoder. This is because we require the encoder to predict the noise from data in order to obtain embeddings, and under the ANM assumption, the mapping from data to noise can be easily expressed as $x\to x-F(x)$ where $F$ is the generative functional mechanism of the generative ANM. However, if we were to consider general SCMs, i.e. of the form $X=F(X, N)$, we would need access to the mapping $x\to F^{-1}(x,\cdot)(x)$ (assuming this function is invertible), which for general functions is not tractable.
>
> Also, it is worth noting that while we require ANM for training the encoder, **we do not need this assumption to train the decoder**. An interesting future work would be to consider a more general dataset encoding (using self-supervised techniques), but we believe this is out of the scope of this work.
>
> > Knowledge of the causal graph for several datasets can potentially be a strong assumption. In real-world datasets, the causal graph may be unknown and must be discovered. However, for the sake of this work, the synthetic scenarios are good for proof of concept.
>
> We thank the reviewer for acknowledging the interest in our synthetic setting. We also agree with the reviewer that the knowledge of true causal graphs might be a strong assumption. However, as explained in the manuscript (l. 234), we can remove this assumption by inferring both causal graphs and functional mechanisms in a zero-shot manner. We have conducted additional experiments showing the extension of our work when the causal graphs are unknown at inference time. To achieve this, we leverage prior work on amortized causal graph learning (AVICI) to predict the graphs in a zero-shot manner from observations only.
>
> **We find that Cond-FiP remains competitive to baselines in this scenario of no access to true causal graph knowledge!** Therefore, our amortized training procedure can be seamlessly adapted for zero-shot inference of both causal graphs and causal mechanisms from observations only. Please refer to the general response above, and Appendix D.1 for more details.

---

> ### Author Response · Authors · 2024-11-22
> **Rebuttal by Authors: Part 2/3**
>
> > Could the authors explain why Cond-FiP performs similarly to some baselines in noise prediction and sample generation, especially when the node scale is the same or smaller than used in training?
>
> First, it is important to note that all baselines utilize the true causal graphs to ensure a fair comparison with Cond-FiP. Additionally, we employ the original implementations of these methods to conduct our experiments. It is also worth mentioning that baselines are trained specifically for each test dataset, thereby representing the gold standard that our zero-shot approach aims to achieve. Remarkably, our approach, although trained on specific scales, demonstrates the ability to generalize to both smaller and, more importantly, larger instance problems, while maintaining comparable performance with other baselines.
>
> Also, we have added new experiments comparing Cond-FiP with other baselines in a scarce data regime. **In such a scenario, we show that Cond-FiP obtains significant improvement over baselines!** Indeed, when the input context size is small we cannot reliably train models from scratch. In contrast, zero-shot inference methods are less impacted as their parameters remain unchanged when given a new dataset, and the inductive bias they have learned during their training phase enables them to generalize even with a smaller input context.  Please refer to the general response above and Appendix D.2 for more details on these experiments.
>
> > How is the FiP model implemented? In the original paper, it seems that the task is the recovery of the topological ordering. Is the FiP baseline here aware of the causal structure of datasets?
>
> We use the original code provided by the authors from the causica python package. In their implementation, the authors provide an adapatation of FiP when the causal graph is known. In our work, we compare Cond-FiP with this variant of FiP for a fair comparison.
>
> > The main capabilities of the proposed framework are noise prediction and observational/interventional sample generation. However, individual counterfactual sample generation is also important in many applications. Can this framework enable counterfactual inference?
>
> Thanks for raising this point. Cond-FiP is indeed able to perform individual counterfactual sample generation. We have added an experiment (Table 22, Appendix D.4) where we compare the counterfactual predictions of our method with the various baselines. We observe that Cond-FiP is slightly worse than the baselines, but competitive for some cases. We leave the improvment of Cond-FiP on counterfactual generation for future work.
>
> > How does the alternating application of transformer blocks E work? Is this just an alternating optimization method where you optimize for samples when nodes are fixed and optimize for nodes when samples are fixed?
>
>
> The alternating block transformer is a forward pass neural network that takes as input a tensor of shape $(B,n,d,d_h)$ where $B$ is the batch size, $n$ is the number of samples, $d$ is the number of nodes and $d_h$ in the hidden dimension of the transformer, and that performs alternatively an attention layer over either the second of the third dimension of the tensor. In practice, we permute the second and third dimensions before applying an attention mechanism to perform attention over the sample dimension, and then permute them back afterward to apply an attention layer over the node dimension. This process closely resembles the recent development of spatio-temporal transformers used for example in "Genie: Generative Interactive Environments".
>
> > For zero-shot inference of the SCM distribution, what is the level of distribution shift that a new dataset can have for this method to be able to extrapolate well?
>
> Thank you for asking this. In our experiments, we followed the setting proposed in AVICI and did not investigate the extrapolation capabilities of Cond-FiP beyond the out-of-distribution datasets presented in the manuscript, that are LIN OUT, RFF OUT, Sachs and C-suite datasets. Since the current model is fairly small ($\simeq 20 M$), it might not be able to generalize for extreme distribution shifts. However, with increased computational resources and additional training data, we hypothesize that our approach could generalize to a broader range of problems. We leave these scaling investigations for future work.

---

> ### Author Response · Authors · 2024-11-22
> **Rebuttal by Authors: Part 3/3**
>
> > In the Adaptive Transformer Block, which iteratively updates the noise conditioned on the dataset embedding $z_{emb}$, can we interpret this as sort of a noise abduction typically performed in counterfactual inference?
>
> The adaptive transformer block can be seen as the generation process that transforms noise to data, while noise abduction refers to the inverse process that transforms data to noise. The confusion might come from the fact that the data is explicitely given at training time. But recall that at inference time, the data is never provided, but rather generated from noise using the predicted functional mechanisms.
>
> > How exactly does one perform interventions in the Cond-FiP? It would be beneficial if the authors elaborate on this mechanism in the paper. From my understanding, we just feed in an intervened SCM causal graph with the mutilations and use the corresponding dataset embedding for conditional generation. However, this is not made very clear in the paper.
>
> Thank you for highlighting this point. We have included detailed explanations on the usage of Cond-FiP at inference time for both generation and interventional prediction tasks in the general response above. We hope that these clarifications adequately address the concerns of the reviewer.

---

> > ### Comment · Reviewer_6M7T · 2024-11-25
> >
> > I greatly appreciate the authors taking the time to provide detailed clarifications and additional results. As I mentioned, I do believe the overall contribution is quite interesting and has a lot of potential for real-world use-cases. The authors have provided some new experiments for counterfactual generation, unknown causal graph setting, and zero-shot evaluation under smaller test sizes, which I believe strengthen the arguments presented in the paper. Although I think there could be some more empirical evaluation for distribution shift robustness, I think the provided experiments show promising results for the applicability of this idea in larger-scale settings for efficient zero-shot SCM inference. Therefore, I raise my score towards the acceptance of this paper.

---

> > > ### Author Response · Authors · 2024-11-26
> > > **Many thanks for reading our rebuttal**
> > >
> > > Dear Reviewer,
> > >
> > >
> > >
> > > We sincerely appreciate your valuable comments and questions, which have greatly contributed to improving our manuscript. We are also truly grateful for the increase in your score.
> > >
> > >
> > >
> > > > Although I think there could be some more empirical evaluation for distribution shift robustness
> > >
> > >
> > >
> > > We plan to assess the robustness of our current trained model by increasing the shift of the out-of-distribution datasets. We look forward to sharing new results with you soon.

---

> ### Author Response · Authors · 2024-11-28
> **Results on Robustness to Distribution Shifts**
>
> We thank the reviewer again for their feedback and engagement with us during the rebuttal! Following their suggestion, we have added results in Appendix D.5 that evaluate how sensitive Cond-FiP is to distribution shifts by comparing its performance across scenarios as the severity of the distribution shift is increased.
>
> To illustrate how we control the magnitude of distribution shift, we discuss the difference in the distribution of causal mechanisms across $P_{IN}$ and $P_{OUT}$ from our main results. The distribution shift arises because the support of the parameters of causal mechanisms changes from $P_{IN}$ to $P_{OUT}$. For example, for the linear causal mechanism case, the weights in  $P_{IN}$ are sampled uniformly from $(-3,-1)\cup(1, 3)$; while in  $P_{OUT}$ they are sampled uniformly from $(-4,-0.5)\cup (0.5, 4)$. We now change the support set of the parameters in $P_{OUT}$ to $(-4\alpha,-0.5)\cup (0.5, 4 \alpha)$, so that increasing $\alpha\geq 1$ makes the distribution shift more severe. We follow this procedure for the support set of all the parameters associated with functional mechanisms and generate distributions ($P_{OUT}(\alpha)$) with varying shift w.r.t $P_{IN}$ by changing $\alpha$. Note that $\alpha=1$ corresponds to the same $P_{OUT}$ as the one used for sampling datasets in our main results.
>
> We conduct two experiments to evaluate the robustness of Cond-FiP to distribution shifts, as described below.
> - **Controlling Shift in Causal Mechanisms.** We start with the parameter configuration of $P_{OUT}$ from the setup in the main results and then control the magnitude of shift by changing the support set of parameters of causal mechanisms.
> - **Controlling Shift in Noise Variables.** We start with the parameter configuration of $P_{OUT}$ from the setup in the main results and then control the magnitude of shift by changing the support set of parameters of the noise distribution.
>
> We find that the performance of Cond-FiP does not change much as we increase $\alpha$, indicating that Cond-FiP is robust to the varying levels of distribution shits in causal mechanisms. However, for the case of controlling shift via noise variables we find that Cond-FiP is quite sensitive: the performance of Cond-FiP degrades with increasing magnitude of the shift ($\alpha$) on the noise for all the tasks.
>
> We provide the tables for sample generation in both cases below for the convenience of the reader, please refer also to Appendix D.5 for more details.
>
> ## Sample Generation under Distribution via Causal Mechanisms
>
> | Total Nodes | Shift Level (α) |  LIN OUT         |  RFF OUT         |
> |-----------|------------------|---------------|---------------|
> | 20          | 1                | 0.07 (0.01)   | 0.30 (0.03)   |
> |          | 2                | 0.06 (0.01)   | 0.34 (0.05)   |
> |          | 5                | 0.06 (0.01)   | 0.35 (0.05)   |
> |          | 10               | 0.06 (0.01)   | 0.29 (0.07)   |
> | 50          | 1                | 0.07 (0.0)    | 0.48 (0.07)   |
> |          | 2                | 0.07 (0.0)    | 0.47 (0.07)   |
> |          | 5                | 0.07 (0.01)   | 0.38 (0.06)   |
> |          | 10               | 0.07 (0.01)   | 0.32 (0.06)   |
> | 100         | 1                | 0.09 (0.01)   | 0.57 (0.07)   |
> |         | 2                | 0.09 (0.01)   | 0.60 (0.05)   |
> |         | 5                | 0.09 (0.01)   | 0.58 (0.05)   |
> |         | 10               | 0.12 (0.02)   | 0.56 (0.06)   |
>
>
> ## Sample Generation under Distribution via Noise Variables
>
> | Total Nodes | Shift Level (α) |  LIN OUT         |  RFF OUT         |
> |-------------|------------------|---------------|---------------|
> | 20          | 1                | 0.07 (0.01)   | 0.30 (0.03)   |
> |          | 2                | 0.07 (0.01)   | 0.45 (0.04)   |
> |          | 5                | 0.07 (0.01)   | 0.59 (0.03)   |
> |          | 10               | 0.07 (0.01)   | 0.58 (0.02)   |
> | 50          | 1                | 0.07 (0.0)    | 0.48 (0.07)   |
> |          | 2                | 0.07 (0.0)    | 0.59 (0.06)   |
> |           | 5                | 0.07 (0.0)    | 0.64 (0.03)   |
> |          | 10               | 0.07 (0.0)    | 0.58 (0.02)   |
> | 100         | 1                | 0.09 (0.01)   | 0.57 (0.07)   |
> |         | 2                | 0.09 (0.01)   | 0.63 (0.05)   |
> |         | 5                | 0.09 (0.01)   | 0.65 (0.03)   |
> |         | 10               | 0.09 (0.01)   | 0.59 (0.02)   |

---

### Author Response · Authors · 2024-11-22
**General Response**

**We thank the reviewers, AC and SAC assigned to this paper for their time and work looking into our submission.**

We thank them in advance for reading our rebuttal and interacting with us for a few more days during the discussion period.

We were happy to see that the paper was overall well received by all 4 reviewers:

- **6M7T:** *This paper is one of the first to consider generalizing the learning [...] of structural causal models [...] and is a significant step toward building causally-aware foundation models.*

- **iSSj:** *The problem is well motivated by recent advances in the area, novel, and of interest to the community.*

- **ADS4:** *The idea of zero-shot learning with conditional FiP is interesting.*

- **gTiX:** *I believe the task proposed by the paper is highly motivated. Additionally, the paper is written in a structured way.*

The most important weaknesses highlighted by reviewers point to:

- Some clarifications on the generation and interventional predictions of Cond-FiP.

&#8594; *We have added detailed explanations on the usage of Cond-FiP at inference time.*

- The knowledge of causal graphs at inference time.

&#8594; *Following the reviewers recommendations, we have conducted new experiments where we first infer in zero-shot the causal graphs from observations only using AVICI [1], and use the predicted graphs in Cond-FiP to infer functional mechanisms.*


- Further empirical investigations of the proposed approach.

&#8594; *We have conducted new experiments where we study the generalization capabilities of Cond-FiP in a scarce data regime, and performed an ablation study of the proposed encoder.*

- Additional precisions on amortized methods.

&#8594; *We have followed the reviewers' suggestions, and provided more background on amortized learning, especially in the context of causality.*

We believe we have addressed all of the points raised by reviewers.

At the moment, all reviewers have scored our paper as good or excellent in presentation. We have received an average 2.75 (3,2,3,3) grade in both soundness and contribution.

We believe that the very supportive comments and detailed scores provided in the reviews are not fully reflected in the overall scores of 6, 5, 5, 5. If the reviewers agree with our assessment, we humbly ask them to raise their score.

---

> ### Author Response · Authors · 2024-11-22
> **Clarifications on Inference using Cond-FiP (Reviewer 6M7T, iSSJ, ADS4)**
>
> **Cond-FiP Model.** Once Cond-FiP is trained, we have access to two trained models: (1) an encoder that given a dataset $D_{\textbf{X}}$ and the causal graph associated $\mathcal{G}$ produced an embedding $\mu(D_{\textbf{X}},\mathcal{G})$ as defined l.277., and (2) a decoder that given an embedding $\mu$ and a causal graph $\mathcal{G}$, produces a function $z\in\mathbb{R}^d\to\mathcal{T}(z,\mu, \mathcal{G})\in\mathbb{R}^d$.
>
> Cond-FiP simply consists of the composition of the two models, that is given a dataset $D_{\textbf{X}}$ and the causal graph associated $\mathcal{G}$, produce a function $z\in\mathbb{R}^d\to\mathcal{T}(z,\mu(D_{\textbf{X}},\mathcal{G}), \mathcal{G})\in\mathbb{R}^d$, which for simplicity we denote $z\in\mathbb{R}^d\to\mathcal{T}(z,D_{\textbf{X}}, \mathcal{G})\in\mathbb{R}^d$. Having clarified this, we can now proceed to more detailed explanations of the inference process using Cond-FiP.
>
> **Data Generation.** Given a dataset $D_X$ and its causal graph $\mathcal{G}$, we denote $z\to\mathcal{T}(z,D_X,\mathcal{G})$ the function infered by Cond-FiP. This function defines the predicted SCM obtained by our model, and we can directly use it to generate new points. More precisely, given a noise sample $\textbf{n}$, we can generate the associated observational sample by solving the following equation in $\textbf{x}$:
>
> $$ \textbf{x} = \mathcal{T}(\textbf{x},D_X,\mathcal{G}) + \textbf{n}$$
>
> To solve this fixed-point equation, we rely on the fact that $\mathcal{G}$ is a DAG, which enables to solve the fixed-point problem using the following simple iterative procedure. Starting with $z_0 =\textbf{n}$, we compute for $\ell=1,\dots,d$ where $d$ is the number of nodes
>
>
> $z_\ell = \mathcal{T}(z_{\ell-1},D_X,\mathcal{G}) + \textbf{n}$
>
>
> After $d$ iteration we obtain that
>
> $z_d = \mathcal{T}(z_d,D_X,\mathcal{G}) + \textbf{n}$
>
> and therefore $z_d$ is the solution of the fixed-point problem above, which corresponds to the observational sample associated to $\textbf{n}$ according to our predicted SCM $z\to\mathcal{T}(z,D_X,\mathcal{G})$.
>
> **Interventional Prediction.** Recall that given a dataset $D_X$ and its causal graph $\mathcal{G}$, $z\in\mathbb{R}^d\to\mathcal{T}(z,D_X,\mathcal{G})\in\mathbb{R}^d$ denotes the SCM infered by Cond-FiP. Let us also denote the coordinate-wise formulation of our SCM defined for any $z\in\mathbb{R}^d$ as $\mathcal{T}(z,D_X,\mathcal{G}) = [[\mathcal{T}(z,D_X,\mathcal{G})]_1,\dots,[\mathcal{T}(z,D_X,\mathcal{G})]_d]$.
>
>
> In order to intervene on this predicted SCM, we simply have to modify in place the predicted function. For example, assume that we want to perform the following intervention $\text{do}(X_i) = a$. Then, in order to obtained the intervened SCM, we need to introduce a new function $z\to\mathcal{T}^{\text{do}(X_i)=a}(z,D_X,\mathcal{G})$ defined for any $z\in\mathbb{R}^d$ as:
> $[\mathcal{T}^{\text{do}(X_i)=a}(z,D_X,\mathcal{G})]_j = [\mathcal{T}(z,D_X,\mathcal{G})]_j$ if $j\neq i$ and $[\mathcal{T}^{\text{do}(X_i)=a}(z,D_X,\mathcal{G})]_i = a$.
>
> Now, using this intervened SCM $z\to\mathcal{T}^{\text{do}(X_i)=a}(z,D_X,\mathcal{G})$, we can apply the exact same generation procedure as the one introduced above to generate intervened samples according to our intervened SCM.

---

> ### Author Response · Authors · 2024-11-22
> **Experiment: On the Knowledge of True Causal Graphs (Reviewer 6M7T, iSSj)**
>
> Prior works in causal learning have proposed new methodologies [1,2] to predict in zero-shot the underlying causal graphs from observational data. Hence, we have conducted new experiments where we do not assume the knowledge of true graphs, but we rather infer them using an amortized causal discovery approach (AVICI [1]) from the observational data. The combined pipeline of AVICI + Cond-FiP enables to infer in zero-shot the full SCMs from oberservations only.
>
> The results for benchmarking Cond-FiP with predicted graphs using AVICI for the task of noise prediction, sample generation, and interventional prediction are provided in Table 10, Table 11, and Table 12 respectively in Appendix D.1. Please find also below the results for the task of sample generation for nodes $d=50$ and $d=100$.
>
> | Method   | Total Nodes | LIN | RIN | LOUT | ROUT |
> |----------|-------------|----------------|----------------|-----------------|-----------------|
> | DoWhy   | 50          | 0.53 (0.07)    | 0.46 (0.06)    | 0.58 (0.03)     | 0.59 (0.07)     |
> | Deci    | 50          | 0.55 (0.07)    | 0.54 (0.07)    | 0.59 (0.02)     | 0.66 (0.06)     |
> | FiP     | 50          | 0.53 (0.07)    | 0.44 (0.05)    | 0.58 (0.02)     | 0.53 (0.07)     |
> | Cond-FiP | 50          | 0.52 (0.07)    | 0.43 (0.05)    | 0.58 (0.02)     | 0.53 (0.07)     |
> | DoWhy   | 100         | 0.67 (0.07)    | 0.52 (0.06)    | 0.69 (0.02)     | 0.68 (0.04)     |
> | Deci    | 100         | 0.69 (0.08)    | 0.57 (0.08)    | 0.69 (0.02)     | 0.71 (0.04)     |
> | FiP     | 100         | 0.66 (0.07)    | 0.50 (0.07)    | 0.68 (0.02)     | 0.64 (0.05)     |
> | Cond-FiP | 100         | 0.64 (0.06)    | 0.49 (0.06)    | 0.68 (0.02)     | 0.63 (0.05)     |
>
>
> For a fair comparison, the baselines FiP, DECI, and DoWhy also use the inferred graphs obtained by AVICI instead of the true ones, and we find that Cond-FiP remains competitive to baselines in this scenario. Therefore, our amortized training procedure can be seamlessly adapted for **zero-shot inference of both causal graphs and causal mechanisms** from observations only.
>
> ### References
>
> [1] Lorch, Lars, Scott Sussex, Jonas Rothfuss, Andreas Krause, and Bernhard Schölkopf. "Amortized inference for causal structure learning." Advances in Neural Information Processing Systems 35 (2022): 13104-13118.
>
> [2] Ke, Nan Rosemary, Silvia Chiappa, Jane Wang, Anirudh Goyal, Jorg Bornschein, Melanie Rey, Theophane Weber, Matthew Botvinic, Michael Mozer, and Danilo Jimenez Rezende. "Learning to induce causal structure." arXiv preprint arXiv:2204.04875 (2022).

---

> ### Author Response · Authors · 2024-11-22
> **Experiment: Benchmark in the Scarce Data Regime (Reviewer 6M7T, gTiX)**
>
> One additional advantage of zero-shot inference methods over traditional is their generalization capabilities when the dataset size becomes smaller. As the sample size in test dataset decreases, the performances of baselines can be severely affected since they require training from scratch on these datasets. In contrast, zero-shot inference methods are less impacted as their parameters remains unchanged when given a new dataset, and the inductive bias they have learned during their training phase enables them to generalize even with a smaller input context.
>
> We empirically verify this by conducting experiments with test datasets of size $n_{\text{test}}=100$ and $n_{\text{test}}=50$, which is smaller than the sample size of $n_{\text{test}}=400$ used in our main experiments (Table 1, 2, 3). The results for experiments with $n_{\text{test}}=100$ are provided in Table 13, 14, 15; and experiments with $n_{\text{test}}=50$ are provided in Table 16, 17, 18 in Appendix D.2.
>
> **We find that Cond-FiP exhibits superior generalization as compared to baselines when the test datasets are smaller.** We report a subset of results here, comparing Cond-FiP with FiP for the task of sample generation.
>
> ### Experiments with $n_{\text{test}}=100$
>
> | Method   | Total Nodes | LIN | RIN | LOUT | ROUT |
> |----------|-------------|----------------|----------------|-----------------|-----------------|
> | FiP     | 10          | 0.13 (0.01)    | 0.29 (0.04)    | 0.18 (0.02)     | 0.15 (0.03)     |
> | Cond-FiP | 10          | 0.09 (0.01)    | 0.20 (0.03)    | 0.09 (0.02)     | 0.14 (0.02)     |
> | FiP     | 20          | 0.17 (0.02)    | 0.34 (0.06)    | 0.17 (0.02)     | 0.39 (0.03)     |
> | Cond-FiP | 20          | 0.08 (0.00)    | 0.31 (0.06)    | 0.13 (0.01)     | 0.37 (0.02)     |
> | FiP     | 50          | 0.13 (0.01)    | 0.38 (0.07)    | 0.14 (0.01)     | 0.58 (0.06)     |
> | Cond-FiP | 50          | 0.10 (0.01)    | 0.32 (0.05)    | 0.12 (0.01)     | 0.54 (0.05)     |
> | FiP     | 100         | 0.15 (0.01)    | 0.40 (0.07)    | 0.19 (0.02)     | 0.67 (0.07)     |
> | Cond-FiP | 100         | 0.11 (0.01)    | 0.35 (0.07)    | 0.14 (0.02)     | 0.63 (0.07)     |
>
> ### Experiments with $n_{\text{test}}=50$
>
>
> | Method   | Total Nodes | LIN | RIN | LOUT | ROUT |
> |----------|-------------|----------------|----------------|-----------------|-----------------|
> | FiP     | 10          | 0.13 (0.03)    | 0.27 (0.03)    | 0.15 (0.02)     | 0.21 (0.03)     |
> | Cond-FiP | 10          | 0.09 (0.01)    | 0.17 (0.01)    | 0.11 (0.01)     | 0.16 (0.01)     |
> | FiP     | 20          | 0.12 (0.01)    | 0.33 (0.04)    | 0.15 (0.02)     | 0.35 (0.04)     |
> | Cond-FiP | 20          | 0.10 (0.01)    | 0.16 (0.01)    | 0.11 (0.01)     | 0.17 (0.01)     |
> | FiP     | 50          | 0.13 (0.01)    | 0.32 (0.03)    | 0.13 (0.01)     | 0.49 (0.05)     |
> | Cond-FiP | 50          | 0.10 (0.01)    | 0.16 (0.00)    | 0.10 (0.01)     | 0.17 (0.01)     |
> | FiP     | 100         | 0.11 (0.01)    | 0.32 (0.04)    | 0.13 (0.01)     | 0.48 (0.02)     |
> | Cond-FiP | 100         | 0.09 (0.01)    | 0.16 (0.01)    | 0.09 (0.01)     | 0.18 (0.01)     |
>
> Hence, in these scarce data regime, Cond-FiP shows significant improvements compared to baselines that require to be trained from scratch and so for each test dataset.

---

> ### Author Response · Authors · 2024-11-22
> **Experiment: Ablation Study of the Encoder (Reviewer ADS4)**
>
> ## Effect of Training Data
>
> We have conducted an ablation study of the encoder considered in Cond-FiP, detailed in Appendix D.3. Our goal here is to measure the effect of the training data on the generalization capabilities of Cond-FiP. We also want to mention that we have performed a similar ablation study of the decoder in Appendix B.
>
> We conduct this ablation study by training two variants of the encoder in Cond-FiP described as follows:
> - Cond-FiP (LIN): We sample SCMs with linear functional relationships during training of the encoder.
> - Cond-FiP (RFF): We sample SCMs with rff functional relationships during training of the encoder.
>
>
> Note that for the training of the subsequent decoder, we sample SCMs with both linear and rff functional relationships as in the main results (Table 1, 2, 3). This ablation helps us to understand whether the strategy of training encoder on mixed functional relationships can bring more generalization to the amortization process.
>
> We present our results of the encoder ablation study for the task of noise prediction, sample generation, and interventional generation in Table 19, 20, and 21 respectively. Our findings indicate that Cond-FiP is robust to the choice of encoder training strategy! Even though the encoder for Cond-FiP (LIN) was only trained on data from linear SCMs, its generalization performance is similar to Cond-FiP where the encoder was trained on data from both linear and non-linear SCMs.
>
> For convenience of the reader we report results for the task of sample generation with $d=50$ and $d=100$ nodes here as well.
>
>
> | Method        | Total Nodes | LIN        | RIN        | LOUT       | ROUT       |
> |---------------|-------------|------------|------------|------------|------------|
> | Cond-FiP (LIN) | 50          | 0.08 (0.01)| 0.26 (0.05)| 0.11 (0.04)| 0.52 (0.08)|
> | Cond-FiP (RFF) | 50          | 0.11 (0.01)| 0.26 (0.05)| 0.15 (0.02)| 0.48 (0.07)|
> | Cond-FiP        | 50          | 0.08 (0.01)| 0.25 (0.05)| 0.07 (0.00)| 0.48 (0.07)|
> | Cond-FiP  (LIN) | 100         | 0.07 (0.01)| 0.27 (0.06)| 0.08 (0.00)| 0.57 (0.07)|
> | Cond-FiP  (RFF) | 100         | 0.11 (0.01)| 0.29 (0.08)| 0.18 (0.03)| 0.61 (0.08)|
> | Cond-FiP        | 100         | 0.07 (0.01)| 0.29 (0.07)| 0.09 (0.01)| 0.57 (0.07)|
>
> ## Effect of Hyperparameters
>
> In the paper we mentioned that the encoder was trained with the learning rate $1e-4$ and batch size of $8$. To evaluate the impact of hyperparameters on encoder training, we experimented with 9 different combinations based on the following grid:
>
> - Learning Rate: $[5e-4, 1e-4, 5e-5]$
> - Batch Size: $[8, 16, 32]$
>
> Our findings indicate that the encoder achieves comparable performance on the validation set across all these hyperparameter configurations. Therefore, our current choice of hyperparameters for training the encoder is well-suited.

---

> ### Author Response · Authors · 2024-11-22
> **Additional Precisions on Amortized Learning (Reviewer ADS4, gTiX)**
>
> Generally speaking, the goal of amortized learning is to learn to predict the solution of similar instances of the same optimization problem. More formally, given some random inputs $\textbf{I}\sim P_\textbf{I}$ and an objective function $L(\theta,\textbf{I})$, the goal of amortized learning is to learn a parameterized model
>
> $M_\phi$ s.t. $ M_\phi (\textbf{I})\simeq \theta^{*}(\textbf{I}):=\text{argmin}_{\theta\in\Theta} L(\theta,\textbf{I})$
>
> To train such a model, amortized learning requires to have access to various pairs
>
> $(\textbf{I},  \theta^{*}(\textbf{I}))$
>
> and optimize the parameters $\phi$ of $M$ in a supervised manner. In the causal literature, this concept has been employed to learn graphs, topological orders, and even the average treatment effect (ATE) from observations. In all these cases, the variable $\textbf{I}$ denotes a dataset of observations $D_{\textbf{X}}$ and the target variable $\theta^{*}(\textbf{I})$ can represent either the causal graph, the topological order or the ATE depending on the task considered.
>
> In our work, we take this technique a step further by directly predicting functions from datasets (and causal graphs). More formally, in our setting, $\textbf{I}:=(D_{\textbf{X}}, \mathcal{G})$ and $\theta^{*}(\textbf{I}):=\textbf{F}$ where $\textbf{F}$ is the true SCM generating $D_{\textbf{X}}$.

---

### Meta-Review · Area_Chair_X1jm · 2024-12-20

**Metareview:**

This paper introduces a zero-shot inference strategy for estimating structural causal models from observational data. The authors claim that this amortized approach can handle both observational and interventional queries on newly seen datasets and can generalize to previously unseen causal mechanisms and graphs.

Although some reviewers recognized the significance of this new direction and appreciated the clarifications and additional experiments provided during the discussion, the paper does not sufficiently intuitively and rigorously explain why the causal structure can be successfully learned in a zero-shot manner.  Additionally, just a few baselines and datasets have been used for comparison. Without a clear explanation of the assumptions and limitations under which the method works, it becomes impractical and unsafe to apply a causal discovery method for real-world datasets.

We believe these concerns are important and need to be clearly addressed, and a major revision is required. For the benefit of this paper, we reject it for now. Please note that this should not be taken as discouragement. We believe that this paper could become a strong contribution after addressing these concerns.

**Additional Comments On Reviewer Discussion:**

During the reviewer discussion, several reviewers initially found the approach intriguing and recognized the novelty of attempting zero-shot inference of SCMs. However, not all concerns were adequately resolved. In particular,  some reviewers struggled to accept that *inferring an entirely new causal generative process can be done reliably in a zero-shot fashion without a stronger theoretical framework.* The authors’ explanations focused primarily on empirical validation and heuristic arguments rather than providing formal guarantees or conditions under which such zero-shot inference would be provably sound.  We believe that this concern is very important and needs to be clearly addressed, and a major revision is required.

---

### Decision · Program_Chairs · 2025-01-22

Reject